# Carotenoid Cleavage Dioxygenase Genes of *Chimonanthus praecox*, *CpCCD7* and *CpCCD8*, Regulate Shoot Branching in *Arabidopsis*

**DOI:** 10.3390/ijms22168750

**Published:** 2021-08-15

**Authors:** Xia Wang, Daofeng Liu, Jie Lin, Ting Zhu, Ning Liu, Ximeng Yang, Jing Ma, Shunzhao Sui

**Affiliations:** Chongqing Engineering Research Center for Floriculture, Key Laboratory of Horticulture Science for Southern Mountainous Regions of Ministry of Education, College of Horticulture and Landscape Architecture, Southwest University, Chongqing 400715, China; wx221069@email.swu.edu.cn (X.W.); liu19830222@163.com (D.L.); lj0215@yorku.ca (J.L.); swuzhuting@email.swu.edu.cn (T.Z.); Ningliu1003@163.com (N.L.); xm2020@email.swu.edu.cn (X.Y.); majing427@swu.edu.cn (J.M.)

**Keywords:** carotenoid cleavage dioxygenase, strigolactones, wintersweet

## Abstract

Strigolactones (SLs) regulate plant shoot development by inhibiting axillary bud growth and branching. However, the role of SLs in wintersweet (*Chimonanthus praecox*) shoot branching remains unknown. Here, we identified and isolated two wintersweet genes, *CCD7* and *CCD8*, involved in the SL biosynthetic pathway. Quantitative real-time PCR revealed that *CpCCD7* and *CpCCD8* were down-regulated in wintersweet during branching. When new shoots were formed, expression levels of *CpCCD7* and *CpCCD8* were almost the same as the control (un-decapitation). *CpCCD7* was expressed in all tissues, with the highest expression in shoot tips and roots, while *CpCCD8* showed the highest expression in roots. Both CpCCD7 and CpCCD8 localized to chloroplasts in *Arabidopsis*. *CpCCD7* and *CpCCD8* overexpression restored the phenotypes of branching mutant *max3-9* and *max4-1*, respectively. *CpCCD7* overexpression reduced the rosette branch number, whereas *CpCCD8* overexpression lines showed no phenotypic differences compared with wild-type plants. Additionally, the expression of *AtBRC1* was significantly up-regulated in transgenic lines, indicating that two *CpCCD* genes functioned similarly to the homologous genes of the Arabidopsis. Overall, our study demonstrates that *CpCCD7* and *CpCCD8* exhibit conserved functions in the CCD pathway, which controls shoot development in wintersweet. This research provides a molecular and theoretical basis for further understanding branch development in wintersweet.

## 1. Introduction

Branching is one of the most important agronomic traits that determine the plant structure and yield. In higher plants, branching depends on the formation of axillary meristems (AMs) and growth of axillary buds [1,2]. Therefore, the degree of branching depends not only on the establishment of AMs but also on their subsequent vitality and growth. The growth of axillary buds is controlled not only by environmental factors (such as phosphorus deficiency) and genetic factors (such as the expression of the signal integrator gene *BRANCHED1*
*[BRC1]* in the bud) but also by plant hormones [1]. Plant hormones act as a hub in a network composed of many regulatory signals during branching [3,4,5].

Auxin inhibits the growth of axillary buds by maintaining apical dominance, while cytokinin (CK) promotes the growth of axillary buds [6,7]. In recent years, strigolactone (SL) has been identified as a new type of endogenous plant hormone that inhibits shoot branching by inhibiting the growth of axillary buds [8,9]. This effect of SLs was discovered in branching mutants defective in SL signaling, including the *ramosus*
*(rms)* mutants of pea (*Pisum sativum*) [10,11,12,13], *more axillary growth (max)* mutants of *Arabidopsis thaliana* [14,15,16], *high tillering dwarf (htd)* and *dwarf (d)* mutants of rice (*Oryza sativa*) [10,17] and *decreased apical dominance (dad)* mutants of petunia (*Petunia hybrida*) [18,19]. The *CAROTENOID CLEAVAGE DIOXYGENASE (CCD)* genes *CCD7* and *CCD8* were, respectively, identified as *DAD1* and *DAD3* in petunia, *MAX3* and *MAX4* in *Arabidopsis**, RMS5* and *RMS1* in pea and *D17* and *D10* in rice [10,14,15,20]. In SL biosynthesis pathway, the carotenoid isomerase *(**D27/AtD27**)* is responsible for transforming all-*trans*-β-carotene into 9-*cis*-β-carotene [21,22]. *CCD7* catalyzes the cleavage of 9-*cis*-β-carotene to form 9-*cis*-β-apo-10′-carotenal, which undergoes CCD8-mediated cleavage and oxygenation to form C18-ketone β-apo-13-carotenone (SL precursor) [21,22,23,24,25]. SLs are then synthesized from their precursor by cytochrome P450 oxygenase (encoded by *MAX1*) [25]. These genes of SLs biosynthesis pathway are necessary for regulating the axillary bud outgrowth and shoot branching in plants.

In 1968, an Australian, C. M. Donald, put forward the idea that plants can maximize the use of light energy, thereby increasing the economic coefficient and crop yield [26]. Intensive research has been conducted on plant-type breeding in cereal crops such as rice, wheat and corn [5,10,17,22,27,28]. Even though annual plants have always been the focus of research on branch development control, perennial woody plants, such as wintersweet (*Chimonanthus praecox*) have the potential for many additional points of regulation in the CCD pathway. The architectural framework of perennial plants depends on many factors, such as pruning, organogenesis of meristems [2]. The exogenous signals from the environment, including extreme temperature and day length, are integrated by the whole plant to influence the process of axillary meristems through dormancy, induction and release stages [29]. To determine the role of the CCD pathway in controlling branch development in the more complex system of perennials, we studied the branch development of wintersweet.

Wintersweet, which belongs to the Calycantaceae family, is a perennial ornamental deciduous shrub, 2.5 to 3.0 m tall, native to China [30]. It is a rare winter flower-viewing plant and has been cultivated for thousands of years [31]. Due to its unique flowering period (late November to March) and strong floral fragrance, it has high ornamental and economic value. In China, wintersweet is widely grown in pots or in gardens for landscaping plants, and is valued for its cut flowers in winter [32]. To date, research on wintersweet has mainly focused on the molecular mechanisms of flower development [31,33], floral scent [34,35], and the regulation of volatile compound [36] and flavonoid biosynthesis [37]. However, research on branch development in wintersweet has been lacking.

The ornamental characteristics and yield of wintersweet cut flowers are affected by branching. However, the role of SL, as an important branch regulating hormone, in branch development in wintersweet remains unknown. To understand the biological functions of the *CCD* gene family in wintersweet, we identified and isolated *CpCCD7* and *CpCCD8* genes from wintersweet. Sequence and phylogenetic analyses revealed that *CpCCD7* and *CpCCD8* are orthologs of *CCD7* and *CCD8*. Expression analysis revealed that *CpCCD7* and *CpCCD8* were down-regulated in wintersweet during branching. *CpCCD7* was mainly expressed in shoot tips, roots and axillary buds, while *CpCCD8* was mainly expressed in roots and axillary buds. Ectopic expression of *CpCCD7* in *Arabidopsis* resulted in the reduction of rosette branches, whereas that of *CpCCD8* had no effect on plant phenotype compared with the control. Overexpression of *CpCCD7* and *CpCCD8* restored the phenotype of the *Arabidopsis* mutants *max3-9* and *max4-1*, respectively. Overall, the results of this study enhance our understanding of the role of *CpCCD7* and *CpCCD8* in the development of lateral branches in wintersweet and provide a basis for exploring the molecular mechanism of branching in wintersweet.

## 2. Results

### 2.1. Cloning and Phylogenetic Analysis of CpCCD7 and CpCCD8

Two *CCD* genes, *CpCCD7* and *CpCCD8*, were isolated from the root samples of wintersweet. The cDNA sequences of *CpCCD7* and *CpCCD8* were obtained from the wintersweet flower transcriptome database [38]. Sequence analysis showed that *CpCCD7* has an open reading frame (ORF) of 1878 bp, which is predicted to encode a 625-amino acid (aa) protein (Appendix A) (GenBank accession MZ351205), with a molecular weight of 69.29 kDa and the theoretical isoelectric point of 8.34. The ORF of *CpCCD8* is 1668 bp, which corresponds to a 625-aa protein (Appendix A) (GenBank accession MZ351206). The predicted molecular weight and theoretical isoelectric point of CpCCD8 are 61.36 kDa and 6.01, respectively.

Amino acid sequence alignments showed that CpCCD7 shares high sequence similarity with its homologs in *Arabidopsis*, petunia, and rice, including AtCCD7, PhCCD7, OsCCD7, respectively. Similarly, CpCCD8 showed high sequence similarity with AtCCD8, PhCCD8 and OsCCD8. Both CpCCD7 and CpCCD8 contained highly conserved four histidine (His) and three glutamic acid (Glu) residues, which determine the substrate-specificity or catalytic activity of CCDs (Figure 1). Phylogenetic analysis showed that *CpCCD7* and *CpCCD8* grouped with the other plant *CCD7* and *CCD8* genes, respectively, in distinct clusters (Figure 2).

### 2.2. Expression Patterns of CpCCD7 and CpCCD8

To analyze the role of *CpCCD7* and *CpCCD8* in shoot branching in wintersweet, we performed quantitative real-time PCR (qRT-PCR) to examine the expression of these genes in the roots of decapitated seedlings without branches at the six-leaf stage (Figure 3A). Root samples were collected at 0 h, 6 h, 3 d, 5 d, 7 d and 9 d after decapitation; roots of seedlings with an intact apical meristem were used as a control. The expression of *CpCCD7* and *CpCCD8* was significantly down-regulated in decapitated seedlings, with the lowest expression at the 6-h time point. At 3 d after decapitation, when the axillary buds began to sprout (Figure 3A,D), expression levels of *CpCCD7* and *CpCCD8* began to increase (Figure 3B,C). At 9 d after decapitation, when new shoots were formed (Figure 3A), expression levels of both genes in decapitated seedlings were similar to those in the control (Figure 3B,C). We also tested the expression of *CpCCD7* and *CpCCD8* in different tissues of wintersweet plants including root, stem, leaf, stem tip, axillary bud and flower (Figure 3E,F). *CpCCD7* expression was detected in all tissues, with the highest expression level in stem tips. The expression level of *CpCCD7* was much higher in stem tips and roots than in other tissues, low expression in stems and leaves (Figure 3E). On the other hand, *CpCCD8* was mainly expressed in roots and axillary buds, with low expression levels in leaves, and almost no expression in stems, stem tips and flowers (Figure 3F). In roots, the transcript level of *CpCCD8* was 1.89-fold higher than that of *CpCCD7*. Therefore, we speculate that SLs inhibit the growth of lateral branches in wintersweet and are mainly synthesized in roots and axillary buds.

### 2.3. Subcellular Localization of CpCCD7 and CpCCD8

To determine the subcellular localization of CpCCD7 and CpCCD8, CpCCD7 and CpCCD8 were fused to the N-terminus of *GFP* (green fluorescent protein) gene. The 35S::*CpCCD7-GFP*, 35S::*CpCCD8-GFP* or 35S::*GFP* (control) construct was transformed into *Arabidopsis* leaf protoplasts. The GFP signal was dispersed throughout the cytoplasm in protoplasts transformed with the control construct but was localized to the chloroplasts in those transformed with the 35S::*CpCCD7-GFP* or 35S::*CpCCD8-GFP* construct (Figure 4). These results indicate that CpCCD7 and CpCCD8 are localized to the chloroplasts.

### 2.4. Effect of CpCCD7 and CpCCD8 Overexpression on the Branching Phenotype of Arabidopsis

To explore the function of *CpCCD7* and *CpCCD8* genes, we generated and transformed *35S::CpCCD7* and *35S::CpCCD8* constructs into wild-type (WT) *Arabidopsis*. A total of 11 and 16 transgenic lines expressing *35S::CpCCD7* and *35S::CpCCD8* constructs, respectively, were generated by hygromycin selection and PCR-based identification. Two homozygous overexpression lines (OE1 and OE2) for each construct were selected for phenotypic analysis (Figure 5B,G). To examine the branching phenotype, we counted the number of rosette branches and stem branches of *CpCCD7-OE* and *CpCCD8-OE* lines grown under long-day (LD) conditions for 35 d. The results showed that the number of rosette branches in *CpCCD7-OE* lines was slightly less than that in WT plants, although the differences were not significant (WT, 2.56 ± 0.1; OE1, 2 ± 0.2; OE2, 2.17 ± 0.2), and the number of stem branches showed no difference between *CpCCD7-OE* lines and WT plants (WT, 5.23 ± 0.3; OE1, 5.18 ± 0.2; OE2, 5.17 ± 0.2) (Figure 5A,C,D). Compared with the WT, *CpCCD8-OE* lines showed no significant difference in the number of rosette branches (WT, 2.2 ± 0.2; OE1, 2.15 ± 0.1; OE2, 2.19 ± 0.2) and stem branches (WT, 5.71 ± 0.3; OE1, 5.70 ± 0.5; OE2, 5.69 ± 0.6) (Figure 5G,I,J).

In many plant species, BRC1 is considered as an important hub for different signals that control the ability of buds to grow [3], and the effect of *MAX* genes on branching could be attributed to the transcriptional control of BRC1 [39]. Therefore, we tested the expression level of *BRC1* in WT and transgenic lines. Compared with the WT, the expression of *AtBRC1* was higher in *CpCCD7-OE* lines (Figure 5E) but similar in *CpCCD8-OE* lines (Figure 5J). This suggests that *CCD7* and *CCD8* inhibit the growth of rosette branches in *Arabidopsis*, which may be attributed to the transcriptional regulation of BRC1.

### 2.5. CpCCD7 and CpCCD8 Genes Restore the Branching Phenotype of Arabidopsis Max Mutants

To further investigate the functions of *CCD7* and *CCD8* genes, the *35S::CpCCD7* and *35S::CpCCD8* constructs were transformed into *Arabidopsis* branching mutants *max3-9* and *max4-1*, respectively. Six *CpCCD7* transgenic lines and five *CpCCD8* complementation lines were obtained. Three independent lines of each construct (Lines 1–3) were selected for phenotypic observation Figure 6F and Figure 7F).

By observing the leaf shape of 14-d-old of WT, *max3-9*, *max4-1* and complementation lines, we found that the petiole length of mutants was shorter than that of the WT (*max3-9*, 5.81 ± 0.1 mm; *max4-1*, 5.79 ± 0.1 mm; WT, 8.32 ± 0.2 mm), whereas that of complementation lines and WT plants showed no significant differences; the petiole lengths of *CpCCD7* complementation lines 1, 2 and 3 were 7.85 ± 0.2, 7.56 ± 0.2 and 7.55 ± 0.2, respectively (Figure 6A,B,D), whereas those of *CpCCD8* complementation lines 1, 2 and 3 were 7.75 ± 0.2, 7.76 ± 0.2 and 7.86 ± 0.2, respectively) (Figure 7A,B,D). Approximately 35 d after transplant, the rosette branches of WT plants, mutants and complementation lines were observed, the number of rosette branches of *max3-9* and *max4-1* mutants were significantly higher (6.13 ± 0.1 and 6.45 ± 0.3, respectively) than of WT plants (1.7 ± 0.2), while the number of rosette branches of complementation lines were similar with WT plants; the number of rosette branches was 1.75 ± 0.2, 1.85 ± 0.3 and 2.15 ± 0.2 in *CpCCD7* complementation lines 1, 2 and 3, respectively (Figure 6C, E), and 2.39 ± 0.2, 3.25 ± 0.3 and 3.55 ± 0.3 in *CpCCD8* complementation lines 1, 2 and 3, respectively (Figure 7C,E). We also examined the expression level of *AtBRC1* in WT, mutant and complementation lines. The results showed that overexpression of both *CpCCD7* and *CpCCD8* up-regulated the expression of *AtBRC**1* in complementation plants (Figure 6G and Figure 7G). Together, these results indicate that the functions of *CpCCD7* and *CpCCD8* in branch development are conserved in wintersweet, and both genes perform their function by regulating the transcription of *BRC1*.

## 3. Discussion

Recently, molecular and genetic studies showed that CCD7 and CCD8 proteins regulate the growth of axillary buds in *Arabidopsis*, petunia and rice through the MAX/RMS/D pathway [14,15,40]. *CCD7* and *CCD8* genes belong to the *CCD* gene family. Only five CCD enzymes have been reported in different plant species: CCD1, CCD4, CCD7, CCD8 and 90-*cis*-expoxycarotenoid cleavage dioxygenases (NCEDs). NCEDs catalyze the rate-limiting step in the abscisic acid (ABA) biosynthetic pathway [41]. CCD1 enzymes cleave linear and cyclic carotenoids produces apocarotenoids involved for flavor and fragrance [42,43]. CCD4 cleaves carotenoids asymmetrically, which contributes to the coloration of plant tissues [44,45,46,47]. However, CCD7 and CCD8 are involved in the synthesis of the precursor of SL from 9-*cis*-β-carotene, namely carlactone. In this study, we isolated two *CCD* genes, *CpCCD7* and *CpCCD8*, from wintersweet. Both these genes play an important role in the development of branches in model plants. Amino acid sequence analysis and structure prediction showed that CpCCD7 and CpCCD8 exhibit the typical characteristics of CCD family proteins. Firstly, four His residues required for binding to the iron cofactor were conserved in CpCCD7 and CpCCD8. In addition, both CpCCD proteins contain three conserved second-shell Glu residues in the active site (Figure 1) [23,48]. These results suggest that CpCCD7 and CpCCD8 employ a similar mechanism to regulate branching in wintersweet as their homologs in other plant species [48,49]. Phylogenetic analysis showed that CpCCD7 and CpCCD8 proteins clustered with the CCD7 and CCD8 groups, respectively (Figure 2). In apple (*Malus domestica*), RNA interference (RNAi) lines of *MdCCD7* showed increased branching [50]. Mutations of the *PhCCD7* or *PhCCD8* gene in petunia led to the loss of branching inhibition [51]. In tomato (*Solanum lycopersicum*), *SlCCD7* has been reported to play roles in multiple processes including SL biosynthesis, shoot branching and carotenoid production [52]. Above of these will provide some references for the functional analysis of *CpCCD7* and *CpCCD8* genes.

In this study, we examined the expression levels of *CpCCD7* and *CpCCD8* during the development of branches in wintersweet seedlings following decapitation. Both *CpCCD7* and *CpCCD8* were significantly down-regulated at 6 h post-decapitation compared with the control, and their expression levels gradually increased with the sprouting and growth of axillary buds (Figure 3B,C). We speculate that CpCCD7 and CpCCD8 inhibit the growth of axillary buds in wintersweet through the MAX/RMS/D pathway. The tissue-specific expression pattern of the *CpCCD*7 gene in wintersweet was different from its homologs in other plant species. For example, among eudicots including of *Arabidopsis*, pea, petunia and tomato, the *MAX3*, *RMS5*, *PhCCD7* and *SlCCD7* genes are mainly expressed in roots and stems, and the expression level of *Arabidopsis MAX3* is 2-fold higher in roots than in stems [15,20,51]. Among monocots, the *Non**-dormant Axillary Bud 1* (*NAB1*) gene of sorghum shows the highest expression level in nodes, followed by stems, roots and young panicles [53]; the *high tillering dwarf*
*1 (htd1)* gene of rice shows strong expression in stems, and the lowest expression level in roots [54]; *ZmCCD7/ZpCCD7* of maize (*Zea mays*) is strongly expressed in roots [27]. Among the *CCD* homologs of perennial woody plants, *PtrMAX3* of poplar (*Populus trichocarpa*) [55], *AcCCD7* of kiwifruit (*Actinidia chinensis*) [2] and *MdCCD7* of apple [50] show the highest expression in roots. In the current study, the *CpCCD7* gene of wintersweet showed the highest expression in stem tips, followed by roots and the lowest expression in stems (Figure 3A). The *CpCCD8* gene of wintersweet was mainly expressed in roots (Figure 3b), similar to its homologous genes, including *Arabidopsis MAX4* [14], petunia *PhCCD8* [56], pea *RMS1* [14], tomato *SlCCD8* [57], tobacco (*Nicotiana tabacum*) *NtCCD8* [58], potato (*Solanum tuberosum*) *StCCD8* [59], poplar *PtrMAX4a* [55] and kiwifruit *AcCCD8* [2]. However, *CCD8* homologs in other plant species show different expression patterns. For example, in maize, *ZmCCD8* shows the highest expression in the shank [60]; rice *D10* is mainly expressed in lateral buds and stem tips, whereas the *D10-like* gene of rice is mainly expressed in panicles [10]. As mentioned above, expression patterns of *CCD7* and *CCD8* differ between in eudicots and monocots. These differences indicate that SLs regulate shoot branching in a species-specific manner [58].

To validate our speculation and to better understand the functions of *CpCCD7* and *CpCCD8*, overexpression lines were obtained. We found that rosette branching was inhibited in *CpCCD7-OE* lines. Even though the difference in the number of rosette branches between the *CpCCD7-OE* lines and WT plants was small, the results were reproducible (Figure 5A,C). On the other hand, no phenotypic differences were detected between the *CpCCD8-OE* lines and WT plants (Figure 5F,H), which was consistent with the results obtained in *Arabidopsis* [61]. It is possible that heterologous expression does not reflect the phenotype of the species of interest, and the complex genetic backgrounds of perennial woody plants and herbaceous plants are different. Alternatively, overexpression of *CpCCD7* or *CpCCD8* alone may not be able to increase the content of SLs in plants. It has been shown that *MAX3* and *MAX4* can act sequentially when cleaving the same carotenoid substrate [62]. Therefore, it is possible that CpCCD7 and CpCCD8 need to be overexpressed together to increase the level of inhibitory compounds, which further reduces the growth of axillary buds [61].

The loss of function of *CCD7* and *CCD8* genes leads to an increase in the number of branches in annuals such as *Arabidopsis* [14], petunia [56], pea [11], rice [10], sorghum [53], tomato [52,57] and potato [59], and in perennial woody plants such as kiwifruit and grapevine (*Vitis vinifera*) [2,63]. Overexpression of kiwifruit genes *AcCCD7* and *AcCCD8* under the control of the cauliflower mosaic virus (CaMV) *35S* promoter in *Arabidopsis* branching mutants restored their branching phenotype [2]. Similarly, overexpression of maize *ZmCCD7/ZpCCD7* in *Arabidopsis max3-9* mutant restored its phenotype [27]. This indicates that the role of the CCD pathway in controlling branch development is conserved across a variety of plant species. In this study, the *CpCCD7* and *CpCCD8* genes also restored the phenotype of *Arabidopsis* mutants *max3-9* (Figure 5) and *max4-1* (Figure 7), respectively. This indicates that the CCD pathway, which controls shoot development in plants, is conserved in wintersweet.

*BRC1* belongs to the *TCP* (TEOSINTE BRANCHED 1, CYCLOIDEA, PROLIFERATING CELL FACTORS) gene family, encodes a key transcription factor that inhibits bud growth, and is the hub of many branch-related mechanisms [3]. *BRC1* was reported for the first time in *Arabidopsis* and pea to act downstream of SL, and the expression of *BRC1* was up-regulated by SL treatment [39,64,65]. The effect of *MAX* on branching could mainly be attributed to the transcriptional control of *BRC1* [39]. We analyzed the relative expression of *AtBRC1* in overexpression lines and restoration mutant lines. Interestingly, the expression level of *AtBRC1* was slightly up-regulated in *CCD7-OE* lines; however, its expression showed no difference between *CCD8-OE* lines and the WT (Figure 5E,J). Additionally, the expression level of *AtBRC1* was up-regulated in both *CCD7* and *CCD8* restoration mutant lines (Figure 6G and Figure 7G). This indicates that *CCD7* and *CCD8* inhibit the growth of axillary buds by up-regulating its downstream gene, *BRC1*, in *Arabidopsis*. Therefore, we conclude that the role of the CCD pathway in controlling branch development is conserved in wintersweet and other species [2,27,58,63].

To date, no study has been reported on the SL biosynthetic genes *CCD7* and *CCD8* in wintersweet. As a woody ornamental plant in winter, the shoot structure of wintersweet plays an important role in ornamental value. The particularity of wintersweet growth and development makes us want to understand the regulation mode of the CCD pathway for branch development. In this study, *CpCCD7* and *CpCCD8* were isolated, and their expression patterns and functional characteristics were analyzed. *CpCCD7* and *CpCCD8* were down-regulated in wintersweet during branching, indicated that they play a negative regulatory role in the axillary bud growth of wintersweet. *CpCCD7* and *CpCCD8* overexpression restored the phenotypes of branching mutant *max3-9* and *max4-1*, and up-regulated the *AtBRC1* gene, respectively. These results indicated that the CCD pathway for branch development of wintersweet was the same as that of the other plants. This study improves our knowledge of *CCD7* and *CCD8* homologous genes in wintersweet and provides a foundation for us, to further research on the molecular regulation mechanism of *CCD7* and *CCD8* genes.

## 4. Materials and Methods

### 4.1. Plant Materials and Growth Conditions

Wintersweet seeds were collected from Southwest University, Chongqing, China. The seeds were soaked in 98% sulfuric acid for 30 min, and then rinsed with flowing water. The surface-sterilized seeds were sown in pots filled with peat: vermiculite mix (3:1, *v*/*v*), and incubated under LD conditions (16-h light/8-h dark) and at a constant temperature of 25 °C [31]. To analyze the expression pattern of *CpCCD7* and *CpCCD8* genes in wintersweet, roots, stems, stem tips and leaves were collected from 2-month-old seedlings, and flowers during the full bloom period were collected from 5-year-old plants. The tissues were flash-frozen in liquid nitrogen and then stored at −80 °C until needed for RNA isolation.

*Arabidopsis max3-9* (SAIL_015785, ABRC stock #: CS9567) and *max4-1* (SAIL_015785, ABRC stock #: CS9568) mutants and wild-type (Columbia-0) plants were used for transgenic experiments. *Arabidopsis* culture conditions were the same as described previously [33].

### 4.2. Cloning of CpCCD7 and CpCCD8 Genes

Total RNA was extracted from the roots of wintersweet seedlings using the EASYspin Plant RNA Rapid Extraction Kit (Aidlab, Beijing, China), following the manufacturer’s instructions. First-strand cDNA was synthesized using the PrimeScript RT reagent Kit with gDNA Eraser (TaKaRa, Dalian, China), following the manufacturer’s instructions. The *CpCCD7* and *CpCCD8* genes were amplified from the root cDNA using Pfu DNA polymerase kit (TransGen, Beijing, China) and sequence-specific primer pairs *CpCCD7*-F/R and *CpCCD**8*-F/R, respectively (Appendix A). The PCR products were cloned into the pMD19-T vector (Takara, Shiga, Japan) for sequencing, as described by Liu et al. [31].

Multiple amino acid sequence alignment was performed using ClustalW with the BioEdit software. The phylogenetic tree was constructed with the MEGA6.0 software under the NJ method with 1000 bootstrap replicates [33]. Amino acid sequences of the CCD proteins of other plants species used in the alignment and phylogenic tree were obtained from National Center for Biotechnology Information (NCBI) (http://www.ncbi.nlm.nih.gov/ accessed on 10 May 2020).

### 4.3. Gene Expression Analysis

To analyze gene expression, qRT-PCR was performed using the SsoFast^TM^ EvaGreen^®^ Supermix and Bio-Rad CFX96 system. The qRT-PCR primers of *CpCCD7* and *CpCCD8* are listed in the Appendix A. All primers were designed using the Primer Premier 6.0 software. The qRT-PCR was performed under the following conditions: 95 °C for 3 min, followed by 40 cycles of 95 °C for 5 s, 60 °C for 5 s and 72 °C for 5 s, and a melt cycle from 65 °C to 95 °C.

Leaves and axillary buds of WT and transgenic plants were collected at 35 d after transplant for *Arabidopsis* qRT-PCR analysis. Root samples of wintersweet were collected at 0 h, 6 h, 3 d, 5 d, 7 d and 9 d after decapitation for wintersweet qRT-PCR analysis. Primers used for qRT-PCR analysis (qRT-*CpCCD7*-F/R, qRT-*CpCCD**8*-F/R and qRT-AtBRC1-F/R) are listed in Appendix A. *CpActin* and *CpTubulin* were used as reference genes for data normalization for wintersweet [33]. The *AtActin* gene (Gene ID: 823805) [39] was used as an internal reference for data normalization for *Arabidopsis* (Appendix A). Three biological replicates were performed for each sample, with each biological sample containing three technical replicates. Gene expression level was analyzed by the 2^−∆∆CT^ method [66].

### 4.4. Subcellular Localization Analysis of CpCCD7 and CpCCD8 Proteins

To determine the subcellular localization of CpCCD7 and CpCCD8, the ORFs of *CpCCD7* and *CpCCD8* without the stop codon were cloned into the pCAMBIA1300 vector using the *Sac*I and *Not*I sites. The resulting plasmids, *35S:CpCCD7-GFP* and *35S:CpCCD8-**GFP*, and the empty vector were separately introduced into *Arabidopsis* protoplasts using *Agrobacterium tumefaciens* strain GV3101 [28]. Protoplast transformation was carried out using the *Arabidopsis* Protoplast Preparation and Transformation Kit (Coolaber, Beijing, China), according to the manufacturer’s instructions, and GFP signal was observed by confocal microscopy (Tokyo, Japan). Primers used for plasmid construction are listed in Appendix A.

### 4.5. Overexpression Plasmid Construction and Arabidopsis Transformation

Coding sequences of *CpCCD7* and *CpCCD8* were cloned into the PGWB551 vector, a binary vector, using the Gateway recombination reactions and sequence-specific primer pairs, *CpCCD7*-F/R and *CpCCD8*-F/R (Appendix A) [27]. The resulting constructs, *35S:CpCCD7-PGWB551* and *35S:CpCCD8-PGWB551*, were introduced into WT and *max Arabidopsis* mutants via the floral dip method [67].

Transgenic lines were selected on MS medium containing 25 µg/mL of hygromycin. Plants were grown in a culture room maintained at LD photoperiod and 22 °C temperature. Homozygous T3 lines were used for phenotypic analysis. The number of rosette branches and stem branches were counted at 35 d after transplant.

### 4.6. Statistical Analysis

Data were statistically analyzed by one-way analysis of variance (ANOVA) and Duncan’s test using the IBM SPSS 22 software (SPSS, Chicago, IL, USA). The values of *p* < 0.05 and *p* < 0.01 were recognized as statistically significant and extremely significant, respectively.

## Figures and Tables

**Figure 1 ijms-22-08750-f001:**
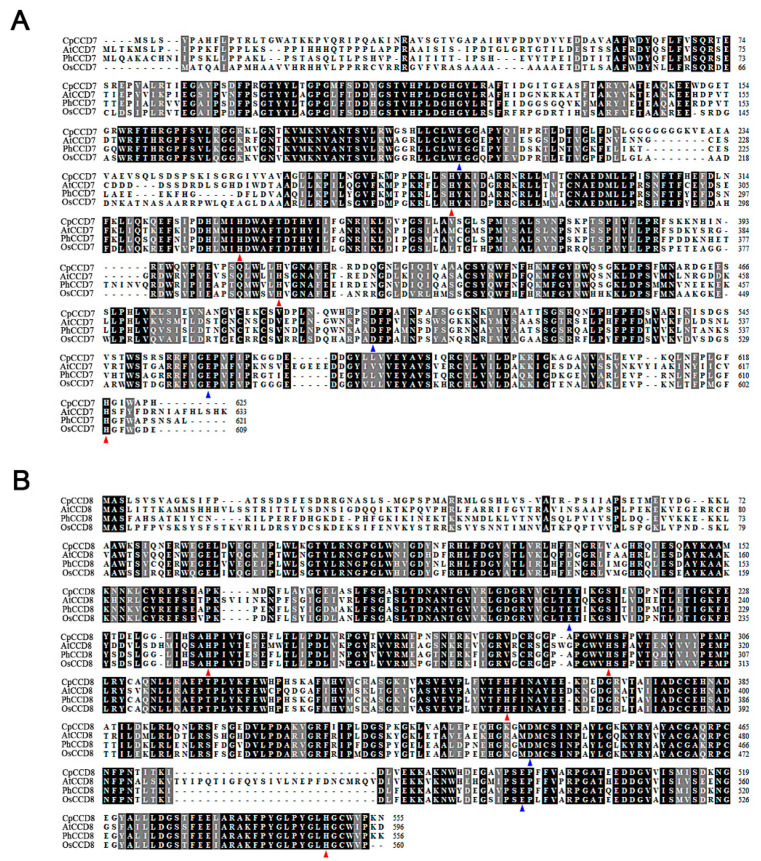
Multiple amino acid sequence alignment of CCD7 and CCD8. (**A**,**B**) Alignments of CCD7 (**A**) and CCD8 (**B**) amino acid sequences of different plant species. Cp, *Chimonanthus praecox*; At, *Arabidopsis thaliana*; Ph, *Petunia hybrid*; Os, *Oryza sativa*. Accessions numbers of various proteins as following: AtCCD7, NP_001324720.1; PhCCD7, ACY01408.1; OsCCD7, AL663000.42; AtCCD8, NP_001329787.1; PhCCD8, AAW33596.1; OsCCD8, XP_015642760.1. Red and blue triangles represent conserved iron-binding His and Glu residues, respectively. Identical amino acids are shaded in black, and similar amino acids were shaded in gray.

**Figure 2 ijms-22-08750-f002:**
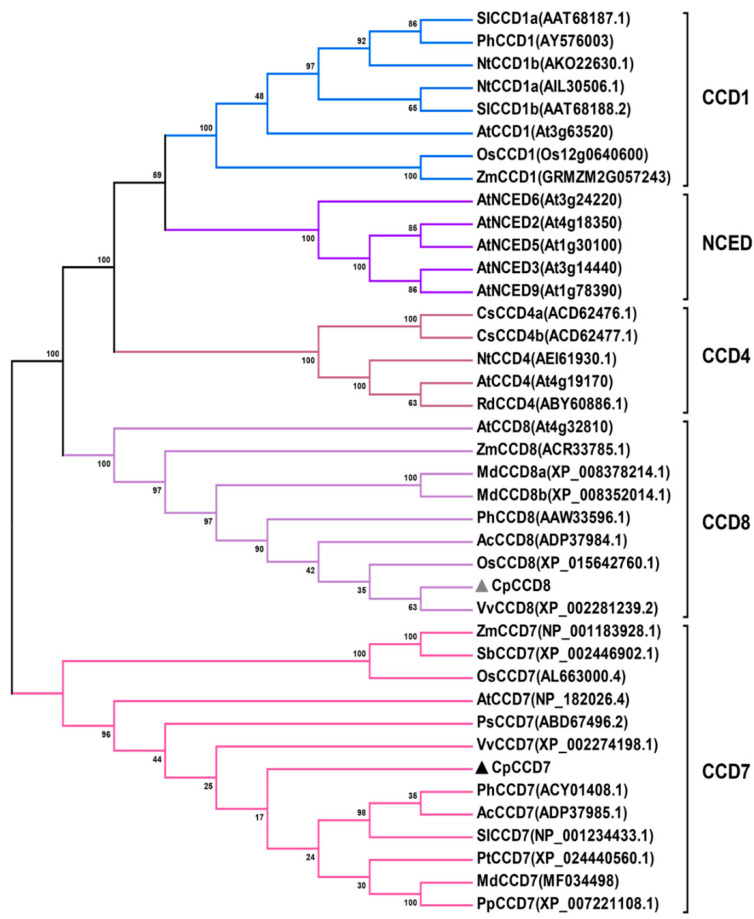
Phylogenetic analysis of CCD family proteins belonging to different plant species. The deduced amino acid sequences of CCD family proteins were aligned using ClustalW in BioEdit Sequence Alignment Editor. Phylogenetic tree was constructed using the neighbor joining (NJ) method, with 1000 bootstrap replicates, using MEGA6.0. At, *Arabidopsis thaliana*; Os, *Oryza sativa*; Ph, *Petunia hybrida*; Zm, *Zea mays*; Nt, *Nicotiana tabacum*; Sl, *Solanum lycopersicum*; Pt, *Populus trichocarpa*; Vv, *Vitis vinifera*; Ac, *Actinidia chinensis*; Pp, *Prunus persica*; Rd, *Rosa damascene*; Cs, *Crocus sativus*; Ps, *Pisum sativum*; Md, *Malus domestica*; Sb, *Sorghum bicolor*. CpCCD7 and CpCCD8 are marked with gray and black triangles, respectively.

**Figure 3 ijms-22-08750-f003:**
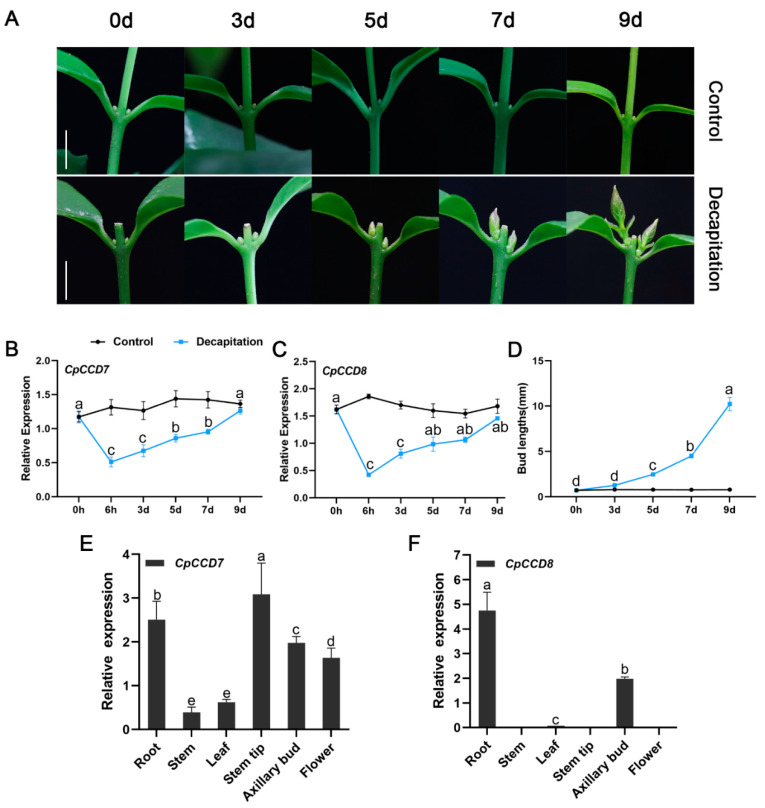
Expression analysis of *CpCCD7* and *CpCCD8* in wintersweet. (**A**) Lateral branch formation process in wintersweet. Scale bar = 1 cm. (**B**,**C**) Expression levels of *CpCCD7* (**B**) and *CpCCD**8* (**C**) in wintersweet during the lateral branch formation process. (**D**) Bud length (mm) after decapitation. (**E**,**F**) Expression levels of *CpCCD**7* (**E**) and *CpCCD**8* (**F**) in different tissues. In (**F**), lack of data in stem, stem tip and flower tissues implies that the expression of *CpCCD**8* was below the detection threshold. Expression levels of *CpCCD7* and *CpCCD8* were normalized to those of *CpAcTin* and *CpTublin*. Data represent mean ± standard error (SE) of three technical replicates. Different lowercase letters (a–e,ab) above bars indicate significant differences (*p* < 0.05).

**Figure 4 ijms-22-08750-f004:**
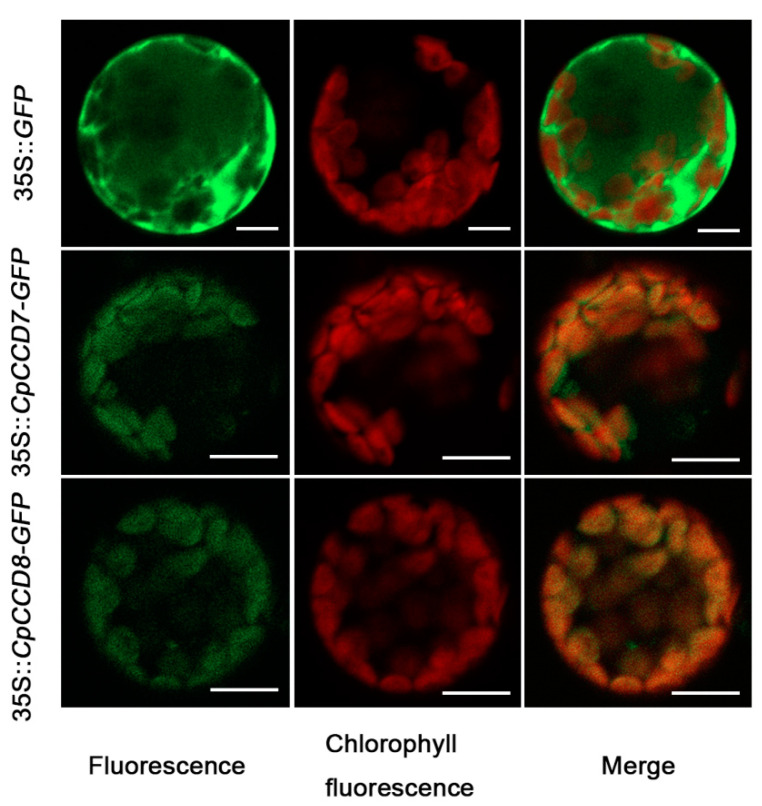
Subcellular localization analysis of GFP-tagged CpCCD7 and CpCCD8. *GFP*-tagged *CpCCD7* and *CpCCD8* genes were expressed in *Arabidopsis* protoplasts. The *35S::GFP* construct was used as the control. Green color indicates *GFP* signal (left panel); red color indicates chlorophyll autofluorescence (middle panel); yellow indicates the merged signal (right panel). Scale bar = 10 µm.

**Figure 5 ijms-22-08750-f005:**
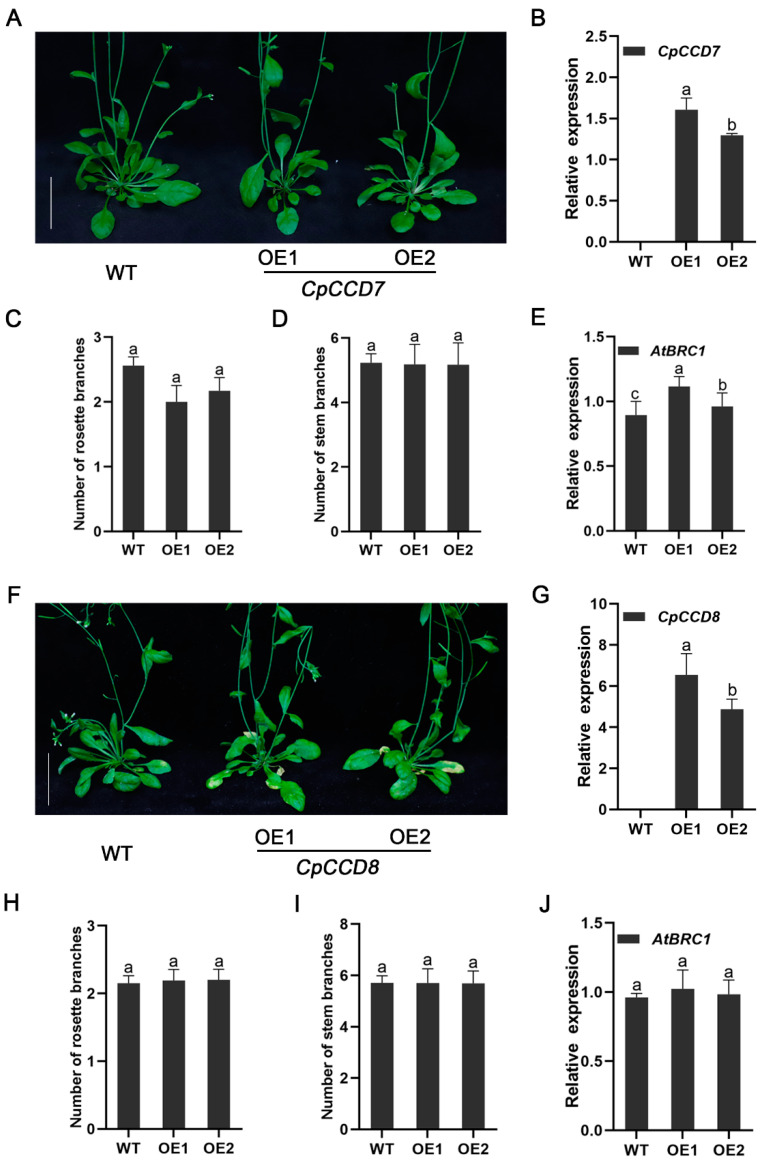
Branching phenotype of *CpCCD7-OE* and *CpCCD8-OE* lines. (**A**) Rosette branching phenotype of *CpCCD7-OE* plants at 35 d after transplant. (**B**) Expression level of *CpCCD7* in transgenic and WT plants. (**C**) Number of rosette branches in *CpCCD7-OE* lines and the WT. (**D**) Number of stem branches in *CpCCD7-OE* lines and the WT. (**E**) Expression level of *AtBRC1* in *CpCCD7-OE* lines and WT. (**F**) Rosette branching phenotype of *CpCCD**8-OE* plants at 35 d after transplant. (**G**) Expression levels of *CpCCD**8* in transgenic and WT plants. (**H**) Number of rosette branches in *CpCCD**8-OE* lines and WT plants. (**I**) Number of stem branches in *CpCCD**8-OE* lines and WT plants. (**J**) Expression level of *AtBRC1* in *CpCCD**8-OE* lines and WT plants. Data represent the mean ± standard deviation (SD; *n* = 20–60). In (**A**,**F**), scale bars = 3 cm. In (**B**–**E**,**G**–**J**), different lowercase letters (a,b) above the bars indicate significant differences (*p* < 0.05).

**Figure 6 ijms-22-08750-f006:**
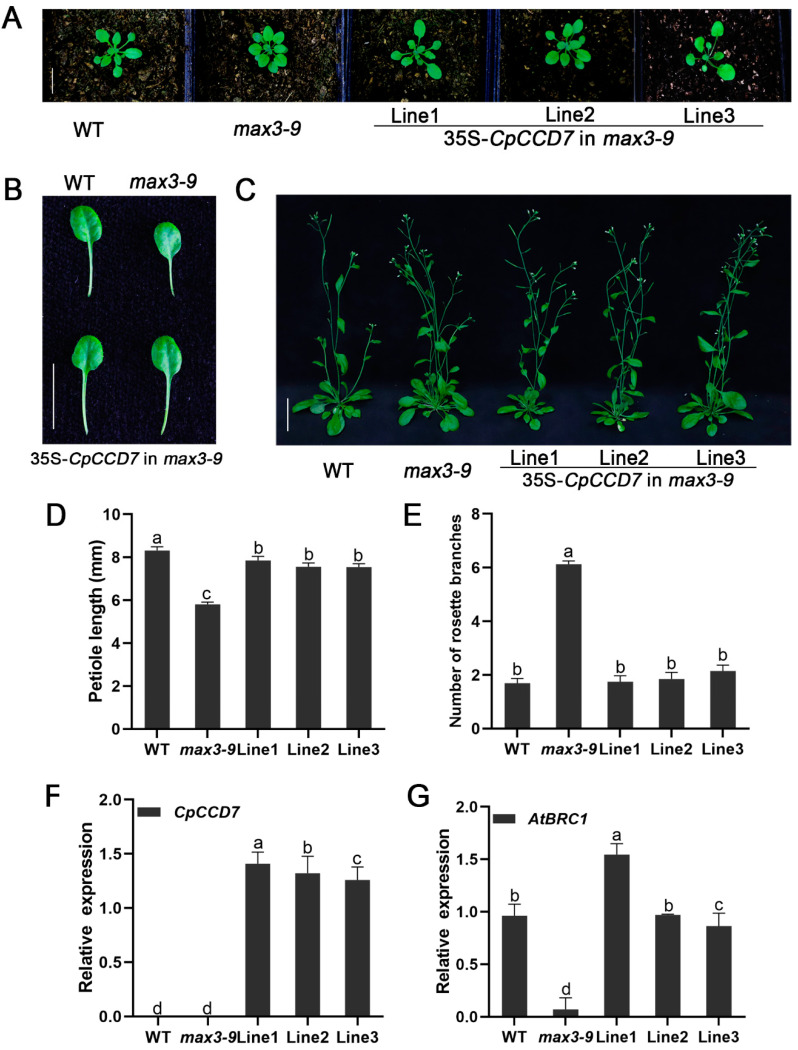
*CpCCD7* overexpression restores the phenotype of the *Arabidopsis* branching mutant *max3-9*. (**A**) Seedlings grown in soil for 2 weeks. (**B**) Petiole phenotype of WT, *max3-9* and restored lines 1–3 grown in soil for 2 weeks. (**C**) Branching phenotype of WT, *max3-9* and complementation lines 1–3 grown 35 d after transplant. In (**A**–**C**), scale bars = 5 mm. (**D**) Petiole length of WT, *max3-9* and plants grown in nutrient-rich soil for 2 weeks. (**E**) Number of rosette branches in WT, *max3-9* mutant and *CpCCD7* complementation lines 1–3. Data represent mean ± SE (*n* = 12–46). (**F**,**G**) Expression levels of *CpCCD**7* and *AtBRC1* in WT, *max**3-9* mutant and *CpCCD**7* restored lines 1–3 (data represent mean ± SD of three biological replicates). The leaves and axillary buds of WT and restored lines were collected used for qRT-PCR analysis at 35 d after transplant. Different lowercase (a–d) above the bars indicates significant differences (*p* < 0.05).

**Figure 7 ijms-22-08750-f007:**
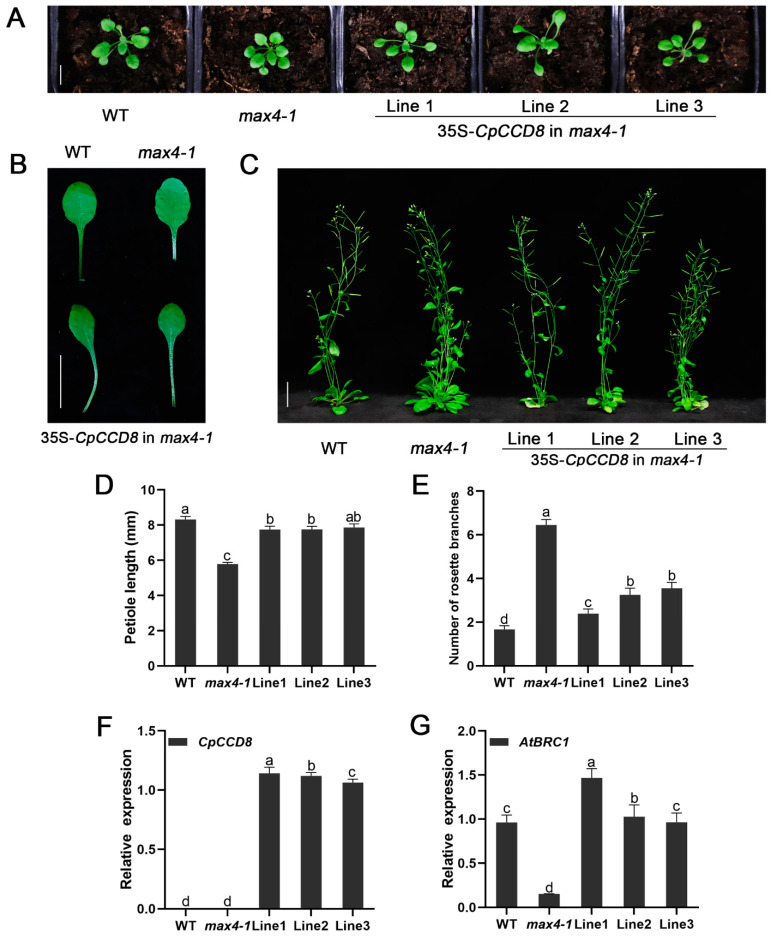
*CpCCD**8* overexpression restores the phenotype of the *Arabidopsis* branching mutant *max4-1*. (**A**) Seedlings grown in soil for 2 weeks. (**B**) Petiole phenotype of WT, *max**4-1* and restored lines 1–3 grown in soil for 2 weeks. (**C**) Branching phenotype of WT, *max**4-1* and restored lines 1–3 at 35 d after transplant. In (**A**–**C**), scale bars = 5 mm. (**D**) Petiole length of WT, *max**4-1* and restored lines 1–3 grown for 2 weeks. (**E**) Number of rosette branches in WT, *max**4-1* mutant and *CpCCD8* restored lines 1–3. Data represent mean ± SE (*n* = 12–46). (**F**,**G**) Expression levels of *CpCCD**8* and *AtBRC1* in WT, *max**4-1* mutant and *CpCCD8* restored lines 1–3 (data represent mean ± SD of three biological replicates). The leaves and axillary buds of WT and complementation lines were collected used for qRT-PCR analysis at 35 d after transplant. Different lowercase (a–d,ab) above the bars indicates significant differences (*p* < 0.05).

## Data Availability

Not applicable.

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
