# Peer review of "Carotenoid Cleavage Dioxygenase Genes of Chimonanthus praecox, CpCCD7 and CpCCD8, Regulate Shoot Branching in Arabidopsis"

_ijms, 2021, doi:10.3390/ijms22168750_

Round 1

Reviewer 1 Report

The manuscript has been significantly improved. I have only few minor comments (see attached file).

Reviewer 2 Report

This paper is written well with good discussion. Conclusions are careful to stay within the scope of inference based on data. The number of OE lines from each construct to assess effects on branching phenotype seems low. The data are well presented and a brief explanation on why only two or three lines were selected for phenotype effect analysis would be helpful.

On Fig 5, 6 & 7 caption, line 228, line 403 (35-d-old), and line 429 (35-day-old), does this refer to 35 days after transformation treatment, transplant, planting, or an approximate development stage? In general, it gives a good idea of growth stage but 'old' seems an unusual word to use.

Author Response

This manuscript is a resubmission of an earlier submission. The following is a list of the peer review reports and author responses from that submission.

Round 1

Reviewer 1 Report

One of the main flaws of the article is that the English grammar is incorrect. The grammatic rules are not being obeyed to. 

ALthough the topic has novelty, the number of experiments done, for this journal is too low. I believe more in depth experiments and functional studies should have been done. 

Besides, the results are poorly described. A very general description is not enough. The most important results must be given in the text, with the values. In Figure 3, part A and B is described as showing the same, but the graphs differ...

Materials and methods are poorly described, the protocols are not given in many instances. If you do not wish to describe the protocol then please give references to the methods used. There are no ID of tested genes given, nor the source of the sequences - databanks name. The same applies for the two genes used for the control in qRT-PCR analyses.

The article is sloppy, in many instances there are too many or not enough spaces, some of the words are written in capital letter with no apparent reason, some sentences start with words written in small letters. Latin names are not written in italics, and common names for plants are... the gene names should be given in full when use for the first time, and all other abbreviations should. 

Many of the sentences are not written properly, meanning the logic is not there.

The discussion has flaws, not only grammatic ones.

Overall, I think that there are too many errors, and as is the article is not suitable for publication.

Detailed comments and necessary corrections are marked in the attached file.

Reviewer 2 Report

The ms describes the isolation of two genes in wintersweet, which are involved in the carotenoid signaling pathway. More specifically, the genes identified showed to be involved in the strigolactone biosynthesis. The Authors build on the knowledge that CCD7 and CCD8 are known, in other species, to be responsible for branching mutants. Their interest focusses on the understanding of the branching pattern of an ornamental species with economic interest. Their results suggest that the two isolated genes - when tested in heterologous system - can restore the phenotype of known branching mutants, but they show a somewhat different response to auxin. 

The work is still quite preliminary, as the Authors correctly state. The conclusions (especially in the Discussion) are too suggestive, as in fact the Authors do not have elements to claim that their findings will help in designing the ideotype for wintesweet.

The ms requires the corrections of a professional editor, including legends, as the mistakes, mispelling, grammar and style slips are too many to be listed here.

The literature presented in the Introduction should be improved citing wintersweet works, also on the pathways studied by the Authors. Also, it would be interesting to discuss the possible nature of the CCD family, given the recent genome assembly. 

On a minor note, the Authors introduce some genes in the Discussion which have not been previously mentioned, and this is a bit confusing.  

In figure 5, "rosette" and "stem" should be added to the y-axes.  

In summary, the work is an average identification of potentially interesting genes, but little is provided in terms of understanding about how they function.